# Post-marketing safety of solriamfetol: A retrospective pharmacovigilance study based on the us food and drug administration adverse event reporting system

Lingling Wu[1,2], Kaijian Zhu(iD)[1,2]*

**1** Department of Pharmacy, Huai'an Hospital Affiliated to Yangzhou University (The Fifth People's Hospital of Huai' an), Huaian, Jiangsu Province, China, **2** Department of Pharmacy, Yangzhou University Huai'an Medical Collaborative Innovation Center, Huaian, Jiangsu Province, China

\* 13515245847@163.com

## Abstract

### Purpose

Excessive daytime sleepiness (EDS) seriously affects quality of life and may increase the risk of life-threatening situations, such as motor vehicle accidents. Solriamfetol is a novel medication approved for the treatment of EDS and serves as an alternative to traditional stimulants. This retrospective pharmacovigilance study aimed to analyze adverse events (AEs) related to solriamfetol based on real-world data.

### Methods

Data regarding solriamfetol-related adverse events were retrieved from the FDA Adverse Event Reporting System (FAERS) from Q3 of 2019 to Q1 of 2024. A total of 1550 reports on solriamfetol-related AEs were analyzed using disproportionality analysis to identify AE signals across various organ systems.

### Results

A large proportion of AEs were reported among female patients (64.06%), primarily including those with narcolepsy (38.13%) and obstructive sleep apnea (3.68%). The most frequently reported AEs included headache, anxiety, and drug ineffectiveness, with 46.8% of AEs occurring within 7 days of treatment initiation. Furthermore, solriamfetol was significantly associated with psychiatric and nervous system disorders as well as cardiac and general disorders.

### Conclusions

Solriamfetol-related adverse events were mainly psychiatric, neurological, cardiac, and general disorders, with headache, anxiety, and drug ineffectiveness being the

**Data availability statement:** The dataset supporting the conclusions of this article is available through the public FDA Adverse Event Reporting System database at https://fis.fda.gov/extensions/FPD-QDE-FAERS/FPD-QDE-FAERS.html.

**Funding:** This work is financially supported by grants from the Jiangsu Pharmaceutical Association (Grant No. H202311).

**Competing interests:** The authors have declared that no competing interests exist.

**Abbreviations:** AEs, adverse events; FAERS, Food and Drug Association Adverse Event Reporting System; FDA, Food and Drug Administration; PTs, preferred terms; SOCs, Social Organ Classes.

most common. Nearly half of the events occurred within the first week of treatment. Given the limitations of the FAERS database, further prospective studies are needed to confirm these findings.

## Introduction

In sleep medicine consultations, individuals often present with excessive daytime sleepiness (EDS), defined as the inability to stay awake during the normal wake period, which may manifest as persistent drowsiness, difficulty staying awake in monotonous situations, automatic behaviors, and impaired sustained attention or vigilance [1,2]. According to recent epidemiological studies, the prevalence of EDS is as high as 33% among adults, and in children and adolescents, the prevalence of EDS is as high as 29.2% [3,4]. EDS significantly reduces the quality of life of patients, leading to decreased productivity, impaired learning, poor concentration, and memory loss [5]. In addition, it has been shown to increase the incidence of several chronic diseases, including cardiovascular events and diabetes [6,7].

The management of EDS involves addressing underlying causes and implementing both non-pharmacologic and pharmacologic strategies. Behavioral interventions such as scheduled napping, sleep hygiene education, and work accommodations may offer partial relief [1]. However, pharmacologic treatment plays a crucial role, particularly for patients with narcolepsy or those with obstructive sleep apnea (OSA) who continue to experience EDS despite effective continuous positive airway pressure therapy. In such cases, medication becomes an essential component of care to improve wakefulness and quality of life [8].

For narcolepsy-related EDS, FDA-approved medications include modafinil, armodafinil, methylphenidate, amphetamine derivatives, sodium oxybate, solriamfetol, and pitolisant [9]. In contrast, pharmacologic options for residual EDS in patients with OSA are more limited, with modafinil, armodafinil and solriamfetol approved in the United States, and only solriamfetol also approved in the European Union for this indication [10]. Moreover, these drugs may have adverse effects, including headache, nausea, dry mouth, anorexia, and diarrhea [9]. In some patients, the use of these drugs alone or in combination may be ineffective [11]. This ineffectiveness can lead to overdose or repeated use of the drugs, resulting in addiction. Therefore, the use of the aforementioned drugs is limited in clinical settings [12].

Solriamfetol was first approved in the United States of America in March 2019 and subsequently approved in the European Union for the treatment of EDS caused by narcolepsy or OSA [13]. Solriamfetol selectively binds to dopamine and norepinephrine transporters and inhibits the reuptake of the two neurotransmitters without promoting monoamine release [14]. It is highly bioavailable, has a weak correlation with eating habits, and can achieve the desired therapeutic effect at a certain dose, which may improve the adherence of patients to treatment [15]. A randomized, double-blind trial of solriamfetol showed no serious adverse events after overdose, demonstrating the safety of the drug [16].

Despite the favorable therapeutic effects of solriamfetol in clinical settings, its side effects cannot be neglected. In a phase 3 clinical trial, the solriamfetol group had a higher incidence of adverse events than the placebo group [17]. Although many clinical trials have investigated solriamfetol-related adverse events, the results require further validation owing to relatively small sample sizes and limited follow-up durations. The FDA Adverse Event Reporting System (FAERS), one of the largest pharmacovigilance databases, assists the FDA in monitoring the safety of drugs and therapeutic products once they have been approved for clinical use [18]. The FAERS database, which is regularly updated, contains authentic reports of drug-related adverse events from various sources, including medical professionals, consumers, and manufacturers. Given the extensive use of solriamfetol worldwide and the limited assessments of its associated AEs, we comprehensively analyzed the system-specific side effects of solriamfetol in this study, aiming to provide a reference for clinical practice.

## Materials and methods

### Data sources and mining

This retrospective pharmacovigilance study was performed using data from the FAERS database from the third quarter of 2019 to the first quarter of 2024. The participant selection methodology is illustrated in Fig 1. The FAERS database collects adverse event reports from healthcare professionals, consumers, and manufacturers. As a spontaneous reporting system, it is subject to several inherent limitations, including reporting bias, underreporting, and the absence of detailed clinical information such as dosage, treatment duration, and patient comorbidities. Additionally, causality between solriamfetol and adverse events cannot be established, as reports are not systematically verified [19]. Given that the FAERS database is based on voluntary reporting, the presence of duplicate entries is likely. Therefore, we screened the raw data and removed all duplicate entries. We selected "solriamfetol" as the target drug and designated it as the primary suspect (PS) in the drug role code within the dataset. This study adhered to the principles of the Declaration of Helsinki and was conducted according to the relevant guidelines. Our study protocol was reviewed and approved by the Research Ethics Committee of Huaian Fifth People's Hospital (No.KY-P-2024-031-01). The consents were waived by the Research Ethics Committee of Huaian Fifth People's Hospital. Since the FAERS database is accessible to the public and patient records are anonymized and de-identified, informed consent are not required for this study. Adverse events (AEs) were coded using the Preferred Terms (PTs) from the Medical Dictionary for Regulatory Activities (MedDRA, version 25.0). AEs and their categories were classified as PTs and System Organ Classes (SOCs), respectively. The time to onset of AEs was defined as the time interval between the initiation of solriamfetol treatment and the occurrence of AEs.

### Statistical analysis and signal detection

Disproportionality analysis is widely used to investigate the potential relationship between drugs and their AEs. In this study, the R software (version 4.1.0) was used to statistically assess disproportionality signals. Four statistical methods, namely, the reporting odds ratio (ROR), proportional reporting ratio (PRR), Bayesian confidence propagation neural network (BCPNN), and multi-item gamma Poisson shrinker (MGPS), were used to identify potential associations between solriamfetol and its AEs based on the disproportionality analysis framework. The specific equations and criteria for these four methods are detailed in S1 Table. These methods were used to detect potential AE signals. A positive signal was considered when the criteria of all four methods were met.

## Results

### Descriptive analysis

From the third quarter of 2019 to the first quarter of 2024, a total of 20,816,909 AEs were reported in the FAERS database. After duplicates and incomplete data were excluded, a total of 1550 reports involving 3012 AEs mentioned

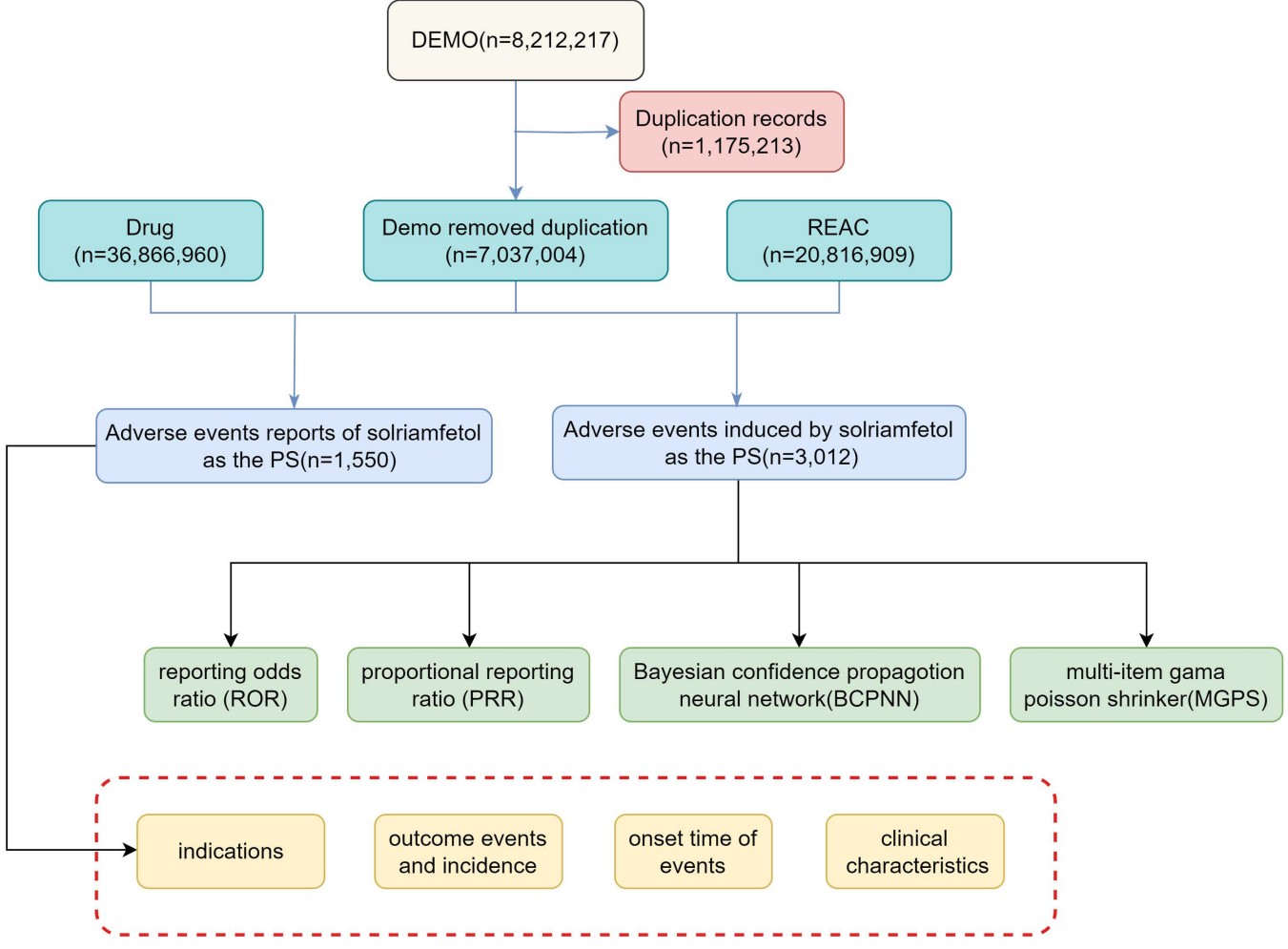

**Fig 1. A flowchart of the participant selection process.**

solriamfetol as the PS. Annual changes in the reported solriamfetol-related AEs are shown in Fig 2, and the clinical characteristics of these AEs are summarized in Table 1. Notably, the proportion of women (993, 64.06%) was higher than that of men (408, 26.32%) in the included AE reports. As shown in Table 1, the most frequently reported indication for solriamfetol was narcolepsy (596, 38.13%), followed by OSA (57, 3.68%). The median age of patients was 40 years; however, age data were unavailable for approximately 78% of cases. Hospitalization (38, 12.97%) was the most frequently reported serious outcome. Most AE reports were from the United States (1403, 90.52%), followed by France (52, 3.35%) and Germany (27, 1.74%). Consumers were the primary reporters (825, 53.23%), followed by physicians (421, 27.16%). In terms of the reporting timeline, the year with the highest number of AE reports was 2022 (549, 35.42%), followed by 2021 (416, 26.84%), 2020 (295, 19.03%), and 2023 (236, 15.23%).

## SOC signals

The signal strengths of AEs at the SOC level are presented in Table 2. The results showed that solriamfetol-induced AEs affected 23 organ systems. The significant SOCs associated with solriamfetol-induced AEs, in which at least one of the four statistical methods (ROR, PRR, BCPNN, and MGPS), included psychiatric disorders (SOC: 10037175); pregnancy,

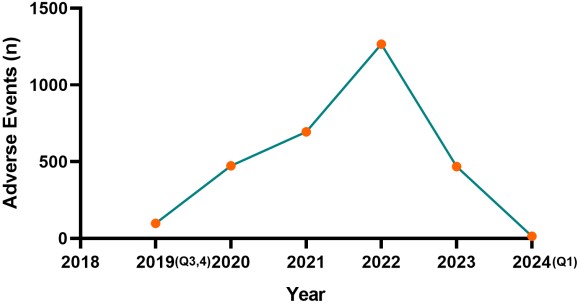

**Fig 2. Number of reported AEs varies by year.** Abbreviations: AE, adverse event.

**Table 1. Basic information of solriamfetol-related AE reports retrieved from the FAERS database (from third quarter of 2019 to the first quarter of 2024).**

| Characteristics of patients | Reports, n (%) |
|---|---|
| Sex | |
| Female | 993 (64.06) |
| Male | 408 (26.32) |
| Unknown | 149 (9.61) |
| Age in years | |
| <18 | 9 (0.58) |
| 18–45 | 198 (12.77) |
| 45–65 | 117 (7.55) |
| ≥65 | 22 (1.42) |
| unknow | 1204 (77.68) |
| Indications | |
| Narcolepsy | 591 (38.13) |
| Obstructive sleep apnea syndrome | 57 (3.68) |
| Unknown | 902 (58.19) |
| Serious outcomes | |
| Hospitalization | 38 (12.97) |
| Death | 5 (1.71) |
| Life-threatening outcomes | 5 (1.71) |
| Disability | 4 (1.37) |
| Other serious outcomes | 241 (82.25) |
| Reporter | |
| Consumer | 825 (53.23) |
| Physician | 421 (27.16) |
| Pharmacist | 242 (15.61) |
| Unknown | 62 (4.00) |
| Reported countries (Top 3) | |
| United States | 1403 (90.52) |
| France | 52 (3.35) |
| Germany | 27 (1.74) |

**Table 2. Signal strengths of solriamfetol-related AEs at the SOC level in the FAERS database.**

| System organ class | N | ROR (95% CI) | PRR (95% CI) | χ2 statistic | IC (IC025) | EBGM (EBGM05) |
|---|---|---|---|---|---|---|
| Psychiatric disorders | 520 | 3.49 (3.18, 3.84) | 3.06 (2.83, 3.31) | 765.34 | 1.61 (1.48) | 3.06 (2.83) |
| Pregnancy, puerperium, and perinatal conditions | 26 | 2.38 (1.62, 3.5) | 2.37 (1.6, 3.51) | 20.57 | 1.24 (0.7) | 2.37 (1.71) |
| Nervous system disorders | 396 | 1.84 (1.66, 2.05) | 1.73 (1.57, 1.91) | 132.85 | 0.79 (0.64) | 1.73 (1.59) |
| General disorders and administration site conditions | 808 | 1.63 (1.5, 1.76) | 1.46 (1.38, 1.55) | 142.58 | 0.54 (0.43) | 1.46 (1.36) |
| Cardiac disorders | 91 | 1.5 (1.22, 1.85) | 1.48 (1.22, 1.8) | 14.59 | 0.57 (0.27) | 1.48 (1.25) |
| Endocrine disorders | 12 | 1.46 (0.83, 2.58) | 1.46 (0.83, 2.58)s | 1.75 | 0.55 (−0.24) | 1.46 (0.91) |
| Immune system disorders | 44 | 1.2 (0.89, 1.62) | 1.2 (0.89, 1.61) | 1.51 | 0.27 (−0.16) | 1.2 (0.94) |
| Investigations | 160 | 0.87 (0.74, 1.02) | 0.88 (0.75, 1.03) | 2.82 | −0.19 (−0.41) | 0.88 (0.77) |
| Ear and labyrinth disorders | 11 | 0.85 (0.47, 1.54) | 0.85 (0.47, 1.53) | 0.28 | −0.23 (−1.05) | 0.85 (0.52) |
| Vascular disorders | 46 | 0.79 (0.59, 1.06) | 0.79 (0.59, 1.06) | 2.54 | −0.33 (−0.75) | 0.79 (0.62) |
| Metabolism and nutrition disorders | 47 | 0.78 (0.58, 1.04) | 0.78 (0.59, 1.03) | 2.87 | −0.35 (−0.76) | 0.78 (0.62) |
| Injury, poisoning, and procedural complications | 301 | 0.77 (0.69, 0.87) | 0.8 (0.73, 0.88) | 18.05 | −0.33 (−0.5) | 0.8 (0.72) |
| Gastrointestinal disorders | 191 | 0.77 (0.66, 0.89) | 0.78 (0.68, 0.89) | 12.8 | −0.36 (−0.57) | 0.78 (0.69) |
| Congenital, familial, and genetic disorders | 6 | 0.7 (0.31, 1.56) | 0.7 (0.31, 1.56) | 0.77 | −0.51 (−1.58) | 0.7 (0.36) |
| Reproductive system and breast disorders | 13 | 0.68 (0.4, 1.18) | 0.69 (0.4, 1.19) | 1.88 | −0.54 (−1.3) | 0.69 (0.43) |
| Respiratory, thoracic, and mediastinal disorders | 85 | 0.59 (0.48, 0.73) | 0.6 (0.48, 0.74) | 23.57 | −0.73 (−1.04) | 0.6 (0.5) |
| Renal and urinary disorders | 31 | 0.5 (0.35, 0.72) | 0.51 (0.36, 0.73) | 15.09 | −0.98 (−1.48) | 0.51 (0.38) |
| Skin and subcutaneous tissue disorders | 83 | 0.45 (0.36, 0.56) | 0.46 (0.37, 0.57) | 54.87 | −1.11 (−1.42) | 0.46 (0.39) |
| Hepatobiliary disorders | 10 | 0.39 (0.21, 0.72) | 0.39 (0.21, 0.73) | 9.6 | −1.36 (−2.21) | 0.39 (0.23) |
| Musculoskeletal and connective tissue disorders | 61 | 0.37 (0.29, 0.47) | 0.38 (0.29, 0.49) | 64.74 | −1.39 (−1.75) | 0.38 (0.31) |
| Eye disorders | 21 | 0.35 (0.23, 0.53) | 0.35 (0.23, 0.54) | 25.61 | −1.51 (−2.11) | 0.35 (0.25) |
| Infections and infestations | 38 | 0.21 (0.15, 0.28) | 0.22 (0.16, 0.3) | 114.26 | −2.21 (−2.66) | 0.22 (0.17) |
| Neoplasms: benign, malignant, and unspecified (including cysts and polyps) | 9 | 0.07 (0.04, 0.13) | 0.07 (0.04, 0.13) | 110.81 | −3.78 (−4.68) | 0.07 (0.04) |

Abbreviations: ROR, Reporting Odds Ratio; PRR, Proportional Reporting Ratio; IC, Information Component; EBGM, Empirical Bayes Geometric Mean.

puerperium, and perinatal conditions (SOC: 10036585); nervous system disorders (SOC: 10029205); general disorders and administration site conditions (SOC: 10018065); and cardiac disorders (SOC: 10007541).

## PT signals

The 30 most common AE signals related to solriamfetol that met the criteria of all four statistical methods (ROR, PRR, BCPNN, and MGPS) are summarized in Table 3. We coded these AEs using PTs from MedDRA® to characterize the toxicity profile of solriamfetol. The results showed that psychiatric and nervous system disorders were reported most frequently in patients treated with solriamfetol.

In particular, the most common AEs were drug ineffective (n = 362, PT: 10013709), headache (n = 143, PT: 10019211), anxiety (n = 120, PT: 10002855), blood pressure increased (n = 56, PT: 10005750), depression (n = 53, PT: 10012378), suicidal ideation (n = 46, PT: 10042458), palpitations (n = 39, PT: 10033557), and insomnia (n = 35, PT: 10022437).

To further explore indication-specific safety profiles, we performed stratified signal analyses based on the reported indications for solriamfetol use. The top AE signals for patients treated for narcolepsy are summarized in S2 Table, while those for patients treated for OSA are shown in S3 Table. While drug ineffective (149 reports in narcolepsy, 14 reports in OSA), headache (54 reports in narcolepsy, 11 reports in OSA), and anxiety (47 reports in narcolepsy, 9 reports in OSA) were the most frequently reported PT in both groups, differences emerged in specific signal strengths. In the OSA

**Table 3. The top 30 AE signals related to solriamfetol ranked by number at the PT level in the FAERS database.**

| Preferred Terms | N | System Organ Class | ROR (95% CI) | PRR (95% CI) | χ2 statistic | IC (IC025) | EBGM (EBGM05) |
|---|---|---|---|---|---|---|---|
| Drug ineffective | 362 | General disorders and administration site conditions | 5.51 (4.93, 6.15) | 4.96 (4.5, 5.47) | 1173.83 | 2.31 (2.15) | 4.96 (4.53) |
| Headache | 143 | Nervous system disorders | 5.14 (4.35, 6.09) | 4.95 (4.23, 5.79) | 454.53 | 2.31 (2.06) | 4.95 (4.3) |
| Anxiety | 120 | Psychiatric disorders | 9.2 (7.67, 11.05) | 8.88 (7.44, 10.59) | 841.38 | 3.15 (2.89) | 8.87 (7.61) |
| Blood pressure increased | 56 | Investigations | 7.17 (5.51, 9.35) | 7.06 (5.47, 9.11) | 291.68 | 2.82 (2.44) | 7.05 (5.65) |
| Depression | 53 | Psychiatric disorders | 5.95 (4.54, 7.81) | 5.86 (4.45, 7.71) | 214.31 | 2.55 (2.16) | 5.86 (4.67) |
| Suicidal ideation | 46 | Psychiatric disorders | 13.4 (10.01, 17.94) | 13.21 (9.85, 17.72) | 518.86 | 3.72 (3.31) | 13.19 (10.33) |
| Palpitations | 39 | Cardiac disorders | 7.93 (5.78, 10.88) | 7.84 (5.73, 10.73) | 232.81 | 2.97 (2.52) | 7.83 (6.01) |
| Insomnia | 35 | Psychiatric disorders | 3.2 (2.29, 4.46) | 3.17 (2.27, 4.42) | 52.26 | 1.67 (1.19) | 3.17 (2.4) |
| Therapeutic response decreased | 35 | General disorders and administration site conditions | 13.27 (9.51, 18.53) | 13.13 (9.41, 18.32) | 391.83 | 3.71 (3.24) | 13.11 (9.92) |
| Migraine | 30 | Nervous system disorders | 6.21 (4.33, 8.9) | 6.16 (4.33, 8.77) | 129.63 | 2.62 (2.11) | 6.15 (4.55) |
| Heart rate increased | 27 | Investigations | 5.91 (4.05, 8.64) | 5.87 (4.04, 8.52) | 109.15 | 2.55 (2.02) | 5.87 (4.27) |
| Feeling jittery | 26 | General disorders and administration site conditions | 41.38 (28.09, 60.94) | 41.03 (27.72, 60.72) | 1009.62 | 5.35 (4.8) | 40.79 (29.5) |
| Exposure during pregnancy | 26 | Injury, poisoning, and procedural complications | 7.69 (5.23, 11.32) | 7.63 (5.16, 11.29) | 149.9 | 2.93 (2.38) | 7.63 (5.52) |
| Hyperhidrosis | 24 | Skin and subcutaneous tissue disorders | 4.51 (3.02, 6.75) | 4.49 (3.03, 6.64) | 65.11 | 2.17 (1.6) | 4.48 (3.2) |
| Sleep apnoea syndrome | 22 | Respiratory, thoracic, and mediastinal disorders | 22.44 (14.74, 34.15) | 22.28 (14.76, 33.63) | 445.89 | 4.47 (3.88) | 22.21 (15.63) |
| Irritability | 22 | Psychiatric disorders | 11.08 (7.28, 16.86) | 11.01 (7.3, 16.62) | 199.96 | 3.46 (2.87) | 10.99 (7.74) |
| Drug ineffective for unapproved indication | 22 | General disorders and administration site conditions | 6.72 (4.42, 10.23) | 6.68 (4.43, 10.08) | 106.28 | 2.74 (2.15) | 6.67 (4.7) |
| Tachycardia | 21 | Cardiac disorders | 5.18 (3.37, 7.95) | 5.15 (3.35, 7.93) | 70.22 | 2.36 (1.76) | 5.14 (3.59) |
| Therapeutic response unexpected | 20 | General disorders and administration site conditions | 10.23 (6.59, 15.88) | 10.17 (6.61, 15.65) | 165.18 | 3.34 (2.72) | 10.15 (7.03) |
| Product administration interrupted | 19 | Injury, poisoning, and procedural complications | 24.89 (15.84, 39.1) | 24.74 (15.76, 38.83) | 431.31 | 4.62 (3.99) | 24.65 (16.89) |
| Therapeutic response shortened | 18 | General disorders and administration site conditions | 7.62 (4.79, 12.11) | 7.58 (4.74, 12.13) | 102.71 | 2.92 (2.27) | 7.57 (5.13) |
| Pre-existing condition improved | 17 | General disorders and administration site conditions | 63.24 (39.18, 102.08) | 62.89 (39.29, 100.66) | 1026.13 | 5.96 (5.29) | 62.33 (41.75) |
| Agitation | 16 | Psychiatric disorders | 6.28 (3.84, 10.26) | 6.25 (3.83, 10.2) | 70.55 | 2.64 (1.95) | 6.24 (4.14) |
| Narcolepsy | 14 | Nervous system disorders | 164.18 (96.52, 279.26) | 163.42 (96.27, 277.42) | 2207.82 | 7.32 (6.58) | 159.67 (102.37) |
| Dry mouth | 14 | Gastrointestinal disorders | 4.12 (2.43, 6.96) | 4.1 (2.42, 6.96) | 32.86 | 2.04 (1.3) | 4.1 (2.64) |
| Disturbance in attention | 12 | Nervous system disorders | 5.19 (2.94, 9.16) | 5.18 (2.93, 9.14) | 40.43 | 2.37 (1.58) | 5.17 (3.22) |
| Panic attack | 9 | Psychiatric disorders | 6.11 (3.17, 11.75) | 6.09 (3.19, 11.63) | 38.29 | 2.61 (1.71) | 6.09 (3.52) |
| Abortion spontaneous | 9 | Pregnancy, puerperium, and perinatal conditions | 5.31 (2.76, 10.22) | 5.3 (2.78, 10.12) | 31.37 | 2.4 (1.51) | 5.29 (3.06) |
| Anger | 8 | Psychiatric disorders | 6.58 (3.29, 13.17) | 6.56 (3.3, 13.03) | 37.72 | 2.71 (1.77) | 6.56 (3.67) |
| Restless legs syndrome | 7 | Nervous system disorders | 8.6 (4.1, 18.07) | 8.59 (4.08, 18.09) | 46.87 | 3.1 (2.1) | 8.58 (4.61) |

Abbreviations: ROR, Reporting Odds Ratio; PRR, Proportional Reporting Ratio; IC, Information Component; EBGM, Empirical Bayes Geometric Mean.

group, several AE signals showed markedly elevated RORs, including therapeutic response unexpected (ROR = 43.56), therapeutic response decreased (ROR = 20.46), and feeling jittery (ROR = 60.88). Cardiovascular-related events such as palpitations (ROR = 6.95) and headache (ROR = 4.85) also ranked high. In contrast, the narcolepsy group exhibited stronger psychiatric-related signals. Notably, suicidal ideation (ROR = 4.54), agitation (ROR = 5.72), mania (ROR = 7.20), and persecutory delusion (ROR = 32.81) were prominent.

### Time to onset of solriamfetol-related AEs

We retrieved the time to onset of solriamfetol-related AEs from the FAERS database (Fig 3). After reports with implausible or missing onset times were excluded, 77 AEs with recorded onset times were retained. The median time to onset was 7 days (interquartile range: 0–55 days). As shown in Fig 3A, a majority of the AEs (n = 36, 46.8%) were reported within 7 days following the initiation of solriamfetol treatment. In addition, the highest number of AEs occurring within 7 days of treatment initiation occurred on the day of the first administration (n = 25, 69.4%) (Fig 3B).

### Discussion

Despite undergoing extensive clinical trials before being approved for commercial use, pharmaceuticals may induce severe AEs in clinical settings. Therefore, post-marketing surveillance is crucial, as it enables prompt identification and management of potential AEs [20].

This study showed notable sex disparity in reports of solriamfetol-related AEs, with 64.06% of reports involving women. These findings suggest that women are more likely to experience or report solriamfetol-related AEs, which may be attributed to physiological differences in drug metabolism between men and women or differences in patient-reported behaviors [21,22]. Among the subset of reports with available age data, the median age of patients experiencing solriamfetol-related AEs was 40 years. While this may suggest that the drug is often used by young and middle-aged adults—consistent with the age distribution of narcolepsy and OSA [23]—the results should be cautious as age data is missing for approximately 78% of cases in our data. Furthermore, the proportion of solriamfetol-related AE reports was higher in the United States (90.52%), which was expected because solriamfetol was first approved for clinical use in the

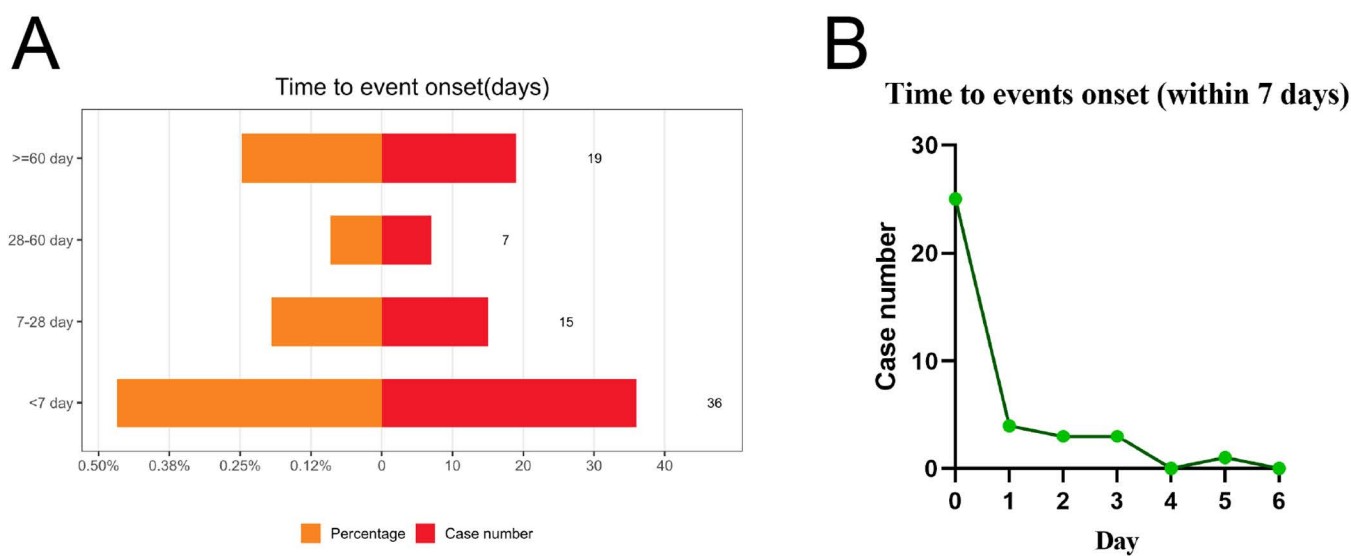

**Fig 3. Time to onset of reported AEs. A.** Time to onset of reported AEs. **B.** Time to onset of reported AEs grouped by days with solriamfetol in the first week. Abbreviations: AE, adverse event.

United States [14]. Additionally, solriamfetol prescriptions in Europe are generally more strictly limited to approved indications (narcolepsy and OSA), whereas off-label use may be more prevalent in the United States. This difference in prescribing restrictions may also contribute to the greater number of AE reports from the United States.

Notably, there was a decline in the number of solriamfetol-related AE reports after 2022, with fewer cases recorded in 2023 and early 2024. Several factors may account for this trend. First, increased familiarity with solriamfetol by clinicians and patients may have led to improved management of expected side effects, thereby reducing the likelihood of reporting. Second, voluntary underreporting may increase over time, especially for known or anticipated AEs. However, due to the passive nature of the FAERS system, the exact reasons behind this decline remain speculative and warrant further exploration.

A total of 30 disproportionality signals were identified for solriamfetol-related AEs, including headache, anxiety, decreased appetite, and insomnia. These signals are consistent with those reported in a previous study [24]. At the SOC level, significant signals were primarily attributed to psychiatric disorders; pregnancy, puerperium, and perinatal conditions; nervous system disorders; general disorders and administration site conditions; and cardiac disorders. It should be noted that the SOC classification strictly follows MedDRA rules and does not necessarily reflect the clinical nature of the events.

General disorders and administration site conditions were the most frequently reported solriamfetol-related AEs. In our dataset, the category 'general disorders and administration site conditions' mainly encompassed adverse events such as drug ineffectiveness, decreased therapeutic response, feeling jittery, and unexpected therapeutic response. In particular, drug ineffectiveness was the most frequent AE. However, a retrospective study showed that 91% of patients on solriamfetol reported a slight or strong improvement in EDS and 94% of physicians found a slight or strong improvement in EDS in their patients [21]. This finding may be attributed to the oral administration of solriamfetol, as its efficacy stabilizes only after 1 week and lasts up to 12 weeks [17]. However, this study showed that a majority of solriamfetol-related AEs (46.8%) were reported within 7 days of treatment initiation and 69.4% of these AEs were reported on the first day of treatment initiation. The drug may not have reached the period of optimal efficacy at the time the AEs were reported. It is also possible that some of these early-onset AEs, occurring predominantly within the first 7 days, may be directly substance-related due to the pharmacological action of solriamfetol. However, non-specific effects such as anticipatory anxiety, placebo–nocebo responses, or adjustment reactions at treatment initiation cannot be excluded and should be considered in interpreting these results. It is also worth noting that the perception of drug ineffectiveness could be influenced by multiple factors such as subtherapeutic dosing, non-adherence, or concomitant medications. However, the FAERS database lacks some detailed and standardized data on dosing regimens, treatment duration, or co-medications, which limits our ability to analyze these influences further.

Among solriamfetol-induced nervous system disorders, headache was one of the most prominent AEs. Consistently, a pooled analysis of three studies showed that the most common adverse reaction in the solriamfetol group versus the placebo group was headache (16% versus 7%) [17,25,26]. Solriamfetol is a dual norepinephrine and dopamine reuptake inhibitor that prevents the resorption of the two neurotransmitters and increases their concentrations in the brain [10]. This inhibition can activate the sympathetic nervous system, leading to vasoconstriction, which may trigger headaches. As a stimulant, solriamfetol promotes wakefulness. Heightened stimulation by solriamfetol may activate pain receptors in the brain, leading to headaches. A similar mechanism is observed for other central nervous system stimulants, such as caffeine, which are known to cause headaches in some individuals [27].

Among solriamfetol-induced psychiatric disorders, anxiety, depression, suicidal ideation, insomnia, irritability, agitation, panic attacks, and anger were more common AEs. Owing to the inhibition of neurotransmitter reuptake, the concentration of neurotransmitters increases in the synaptic gap. This increase in the levels of neurotransmitters promotes wakefulness and concentration while potentially triggering excessive nervous system arousal, which can lead to mental abnormalities such as anxiety, agitation, and nervousness. Abnormalities in the nervous system typically lead to severe AEs. In a study

investigating the long-term safety of solriamfetol, approximately 10% of patients discontinued treatment owing to severe AEs and the most common AEs leading to discontinuation were psychiatric disorders [28]. When participants with a history of psychiatric disorders were excluded, no AE signals indicative of psychiatric disorders were detected [29]. These findings emphasize that a detailed psychiatric and medical history assessment is necessary before prescribing solriamfetol. Treatment may be initiated with the lowest effective dose of solriamfetol, and the dose should be increased gradually to minimize the risk of psychiatric side effects.

Among solriamfetol-induced cardiac disorders, palpitations and tachycardia were the most common AEs. By increasing norepinephrine and dopamine levels, solriamfetol stimulates the sympathetic nervous system, leading to enhanced cardiovascular responses such as an increased heart rate and contractility. This state of sympathetic arousal can manifest as palpitations and increased blood pressure [30]. In the EU, solriamfetol is contraindicated in patients with recent myocardial infarction, serious arrhythmias, or other severe cardiac conditions [31]. In the United States, solriamfetol is not recommended for patients with unstable or severe cardiovascular disease; however, contraindication is not deemed mandatory [32]. Altogether, clinicians should assess the heart rate and blood pressure of patients before initiating treatment with solriamfetol and periodically during treatment to obtain better insights into the safety of the drug.

With regard to the effects of solriamfetol on pregnant women, 9 cases of spontaneous abortion were reported. This is of particular concern, especially for women of childbearing potential. In the EU, solriamfetol is not recommended during pregnancy [31]. However, in the United States, evidence regarding the occurrence of solriamfetol-related AEs in pregnant women is inadequate [32]. Given the potential severity of this adverse event, solriamfetol should be prescribed to women of reproductive age only after careful risk–benefit evaluation. Pregnancy prevention strategies should be discussed and implemented where appropriate.

When comparing our findings with the approved U.S. prescribing information for solriamfetol, several newly detected AEs emerged from the FAERS database that are not prominently highlighted in the official labeling. Specifically, unexpected therapeutic response, shortened therapeutic response, irritability, agitation, and suicidal ideation showed statistically significant signals across all four disproportionality algorithms. While common AEs such as headache, anxiety, nausea, and insomnia are consistent with those listed in the prescribing information, the identification of psychiatric symptoms like persecutory delusion and mania—particularly among narcolepsy patients—suggests the need for heightened clinical awareness. Moreover, cardiovascular-related events such as palpitations and increased blood pressure also demonstrated elevated signal strength, especially in the OSA group, and warrant further prospective evaluation. These findings underscore the added value of post-marketing surveillance in detecting rare or unexpected AEs that may not have been captured during pre-approval trials.

The FAERS database contains sufficient reports to identify rare adverse reactions that are difficult to detect through traditional epidemiological methods. However, it has several inherent limitations. First, as a global spontaneous reporting system, the FAERS database is subject to inherent selection biases, including incomplete documentation of reported cases. Additionally, as voluntary reporting allows consumers to submit AE reports, the professionalism of some reports may be compromised. Second, controversy may arise regarding certain AEs because the FDA does not mandate evidence of causation. Consequently, we could not establish a causal relationship between solriamfetol and the reported AEs. Third, the FAERS database lacks comprehensive information; therefore, confounding factors such as age, comorbidities, and other variables were not controlled for in this study. Fourth, owing to the limitations of the FAERS database, we did not conduct correlation analysis on data collected from the same locations or countries, which might have introduced bias. Fifth, most AEs were reported in patients with narcolepsy, a population in which polypharmacy is common due to the complex symptomatology and need for combination treatment. However, the FAERS database does not consistently provide data on co-administered medications, which limits our ability to account for confounding factors. Despite the significant limitations of the FAERS database for pharmacovigilance studies, we identified and analyzed AE signals related to solriamfetol that may guide future clinical studies. Altogether, the ongoing monitoring of the efficacy and safety of solriamfetol remains critical.

## Conclusion

In conclusion, our analysis of FAERS data provides real-world pharmacovigilance insights into the adverse event profile of solriamfetol. The most frequently reported side effects—including headache, anxiety, and insomnia—are consistent with those observed in prior clinical trials. However, due to the limitations inherent to spontaneous reporting systems, including potential bias, incomplete data, and lack of causality confirmation, these findings should be interpreted with caution. Further prospective studies are warranted to validate these associations.

## Supporting information

**S1 Table. Four major algorithms used for signal detection.**
(DOCX)

**S2 Table. Signal Strength of Adverse Events Associated with Solriamfetol for the Treatment of Narcolepsy: Ranked by Number of Reports at the PT Level in the FAERS Database.**
(DOCX)

**S3 Table. Signal Strength of Adverse Events Associated with Solriamfetol for the Treatment of Obstructive Sleep Apnea: Ranked by Number of Reports at the PT Level in the FAERS Database.**
(DOCX)

## Acknowledgments

We thank Bullet Edits Limited for the linguistic editing and proofreading of the manuscript.

## Author contributions

**Conceptualization:** Kaijian Zhu.

**Formal analysis:** Lingling Wu.

**Methodology:** Kaijian Zhu.

**Writing – original draft:** Lingling Wu.

**Writing – review & editing:** Lingling Wu.

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
