## [Decision Letter · Decision Letter 0]

6 May 2025

PONE-D-25-14184Post-marketing Safety of Solriamfetol: A Retrospective Pharmacovigilance Study Based on the US Food and Drug Administration Adverse Event Reporting System

PLOS ONE

Dear Dr. Zhu,

Thank you for submitting your manuscript to PLOS ONE. After careful consideration, we feel that it has merit but does not fully meet PLOS ONE’s publication criteria as it currently stands. Therefore, we invite you to submit a revised version of the manuscript that addresses the points raised during the review process.

We look forward to receiving your revised manuscript.

Kind regards,

Christian Veauthier, M.D.

Academic Editor

PLOS ONE

 [This work is financially supported by grants from the Jiangsu Pharmaceutical Association (Grant No. H202311)]. 

3. In the online submission form, you indicated that [The data sets generated during the present study are available from the corresponding author upon reasonable request.].

Additional Editor Comments (if provided):

Reviewers' comments:

Reviewer's Responses to Questions

**Comments to the Author**

1. Is the manuscript technically sound, and do the data support the conclusions?

Reviewer #1: Partly

Reviewer #2: No

Reviewer #3: Yes

2. Has the statistical analysis been performed appropriately and rigorously? 

Reviewer #1: Yes

Reviewer #2: Yes

Reviewer #3: Yes

3. Have the authors made all data underlying the findings in their manuscript fully available?

Reviewer #1: Yes

Reviewer #2: Yes

Reviewer #3: Yes

4. Is the manuscript presented in an intelligible fashion and written in standard English?

Reviewer #1: Yes

Reviewer #2: Yes

Reviewer #3: Yes

5. Review Comments to the Author

Reviewer #1: The study analizes AEs of solriamfetol retrospectively using real-world data.

P4 line 55: Authors should clearly state if they refer to EDS in general, in narcolepsy or in OSA. Otherwise the mentioning of specific wake promoting agents does not make sense, since they are approved for certain indications only. Pitolisant as alternative has not been mentioned.

Table 1: it should be mentioned that the prescription in Europe is much more restricted to the approved indications as in the US, which could attribute to the high amount of AEs in the US.

Table 2 needs a legend of abbreviations. The AEs are rather unspecfic. There seems to be much more information about different psychiatric disorders and cardiac symptoms as can be seen in table 3. Could drug ineffectiveness be dependant on dosage or comedication? There is no information about this. Therefore it would be nice to have some more information about the settings of the FAERS database.

What are general disorders, please explain in more detail.

What do you consider to be poisoning? Is this term directly related to intake of solriamfetol only or is it including all preexisting forms of medication, alcohol and drug abuse etc.?

I suggest to give table 3 a better structure by listing all psychiatric disorders first according to frequency followed by nervous system disorders etc.

P19 line 163: Since most AEs were reported within the first 7 days it would be good to discuss if these were in fact substance related or unspecific. Moreover the most important AE for women would be spontaneous abortion. This needs special consideration, because it would imply not to prescribe solriamfetol for women with child bearing potential.

Given all the limitation of the FAERS database at the end of the discussion, the conclusion should be much more cautious. It should mention that the most common side effets of the initial solriamfetol studies were confirmed. I suggest to mention the study limitations clearly in the method chapter to shed a light on the presented results, instead of presenting them at the very end.

The discussion should certainly comment the decline of AEs from 2023 on.

Reviewer #2: The paper by Wu and Zhu describes the frequency of AEs due to Solriamfetol treatment. They used the FEARS database for analysis.

The paper is overall well structured and written.

The paper has many limitations and severe concerns that need to be addressed:

1.- Funding: Please elaborate further on the funding of this project. I could not find any additional information on the Grant No. H202311.

2.- Abstract: “EDS…and may be life-threatening”. This is true in the context of car accident etc., but the wording here is a little unfortunate. Please rephrase this sentence.

3.- Introduction: For a better description of what is EDS you should consider Lammers et al., Sleep Med Rev, 2020.

4.- Introduction: I disagree with your statement “Although continuous positive airway pressure is effective, EDS may persist in patients with OSA, almost all of whom undergo pharmacological therapy [8]”. Please provide data on this statement. I´m not aware of such data. Only some patients undergo pharmacological therapy, not “almost all”.

5.- Introduction: Next, you describe “Modafinil and armodafinil promote arousal and are currently used as first-line treatments for EDS in adults”. The reference you provide does not refer to EDS in OSA or in narcolepsy. You´re not precise here and are mixing up different things. Please clarify in the introduction what are current pharmacological recommendations for EDS in narcolepsy, for EDS in OSA and for other, if available.

6.- Methods/ Discussion/ Conclusions: Limitation of FEARS database: It includes data from “various sources, including medical professionals, consumers,…”. And as described in the results, mostly consumers (>50%) reported AEs. The FEARS database itself describes all its limitations on their webpage. You mention most, but not all of them. Further, for a lot of data “unknown” or “other” is described. I.e. you cannot make strong statement on age “median age was 40”, if 78% of data on age are “unknown”. This also refers to other points in your discussion.

7.- The discussion is too long and repeats results.

8.- Conclusion: I disagree with your conclusion. It is very worrying that, despite the very limited data available and all the serious limitations, you end up concluding that “serious AEs” occur with solriamfetol. The side effects mentioned generally correspond to the known side effects and no real statement can be made about the frequency due to the given limitations. What are the serious AEs?

9.- Most AEs are reported in the context of treatment for narcolepsy. This is not mentioned or discussed. Many patients with narcolepsy receive co- medication or polypharmacy. Another limitation, which is not mentioned.

10.- As further recommendation: In my opinion, it would be interesting if you could split the data sets into a part with results from consumer reports and a part with reports from physicians and compare results.

Reviewer #3: This paper summarizes the postmarketing reports of adverse effects of solriamfetol. The methodology and statistical analysis is sound, the manuscript is written well and the discussion is appropriate, addressing the relevant imitations of such methodology.

The only point I am curious about is the details of serious adverse effects such as hospitalization and death. If possible, it would be very enlightening to add the details regarding these serious AEs, like why the patients were hospitalized or left disabled etc.

Other than that, I have no further comments. It was a pleasure to review this paper.

Thank you.

6. PLOS authors have the option to publish the peer review history of their article (what does this mean? ). If published, this will include your full peer review and any attached files.

**Do you want your identity to be public for this peer review?** For information about this choice, including consent withdrawal, please see our Privacy Policy .

Reviewer #1: **Yes: ** Geert Mayer

Reviewer #2: No

Reviewer #3: No

---

## [Author Response · Author response to Decision Letter 1]

5 Jun 2025

We revised the article and responded to the reviewer in the responses letter, thank you for your help

---

## [Decision Letter · Decision Letter 1]

1 Jul 2025

PONE-D-25-14184R1Post-marketing Safety of Solriamfetol: A Retrospective Pharmacovigilance Study Based on the US Food and Drug Administration Adverse Event Reporting SystemPLOS ONE

Dear Dr. Zhu,

Thank you for submitting your manuscript to PLOS ONE. After careful consideration, we feel that it has merit but does not fully meet PLOS ONE’s publication criteria as it currently stands. Therefore, we invite you to submit a revised version of the manuscript that addresses the points raised during the review process. ==============================

You have addressed the critical comments of the experts; unfortunately, methodological and substantive weaknesses continued to emerge during the review, which I would like to explain:

On the one hand, some reviewers criticized the fact that the data are not new and does not add any significant information to the data already published, but this is not a criterion for publication in Plos One.

Secondly, and more important, after inviting additional reviewers, there are also points of content that make publication in the current version impossible. The basic problem here seems to be that You are writing an article about a molecule (Solriamfetol) and not about a therapy for a distinct disorder. The situation would be completely different if the side effects of treating narcolepsy patients with Solriamfetol had been evaluated – or OSA patients with Solriamfetol…. But different indications are being lumped together here. Moreover, there seem to be also patients without narcolepsy or OSA? And there is little information about comorbidities. But the effects can only be meaningfully assessed if the indication and comorbidities are also known. Otherwise, this should have to be described as a serious methodological limitation (e.g. how many patients with elevated blood pressure already had elevated blood pressure before or were taking antihypertensives? Did this only occur in sleep apnea patients or also in narcolepsy patients?).

Therefore, before publication in Plos One, a fundamental major revision of the article would be necessary, highlighting the methodological shortcomings mentioned, and we cannot guarantee that the article will be accepted next time. In addition, I also refer to further comments from the reviewers.

We look forward to receiving your revised manuscript.

Kind regards,

Christian Veauthier, M.D.

Academic Editor

PLOS ONE

**Additional Editor Comments:**

You have addressed the critical comments of the experts; unfortunately, methodological and substantive weaknesses continued to emerge during the review, which I would like to explain:

On the one hand, some reviewers criticized the fact that the data are not new and does not add any significant information to the data already published, but this is not a criterion for publication in Plos One.

Secondly, and more important, after inviting additional reviewers, there are also points of content that make publication in the current version impossible. The basic problem here seems to be that You are writing an article about a molecule (Solriamfetol) and not about a therapy for a distinct disorder. The situation would be completely different if the side effects of treating narcolepsy patients with Solriamfetol had been evaluated – or OSA patients with Solriamfetol…. But different indications are being lumped together here. Moreover, there seem to be also patients without narcolepsy or OSA? And there is little information about comorbidities. But the effects can only be meaningfully assessed if the indication and comorbidities are also known. Otherwise, this should have to be described as a serious methodological limitation (e.g. how many patients with elevated blood pressure already had elevated blood pressure before or were taking antihypertensives? Did this only occur in sleep apnea patients or also in narcolepsy patients?).

Therefore, before publication in Plos One, a fundamental major revision of the article would be necessary, highlighting the methodological shortcomings mentioned, and we cannot guarantee that the article will be accepted next time. In addition, I also refer to further comments from the reviewers.

Reviewers' comments:

Reviewer's Responses to Questions

**Comments to the Author**

1. If the authors have adequately addressed your comments raised in a previous round of review and you feel that this manuscript is now acceptable for publication, you may indicate that here to bypass the “Comments to the Author” section, enter your conflict of interest statement in the “Confidential to Editor” section, and submit your "Accept" recommendation.

Reviewer #1: All comments have been addressed

Reviewer #3: All comments have been addressed

Reviewer #4: All comments have been addressed

Reviewer #5: (No Response)

2. Is the manuscript technically sound, and do the data support the conclusions?

Reviewer #1: Partly

Reviewer #3: Yes

Reviewer #4: Yes

Reviewer #5: Partly

3. Has the statistical analysis been performed appropriately and rigorously? 

Reviewer #1: N/A

Reviewer #3: Yes

Reviewer #4: Yes

Reviewer #5: Yes

4. Have the authors made all data underlying the findings in their manuscript fully available?

Reviewer #1: Yes

Reviewer #3: Yes

Reviewer #4: Yes

Reviewer #5: Yes

5. Is the manuscript presented in an intelligible fashion and written in standard English?

Reviewer #1: Yes

Reviewer #3: Yes

Reviewer #4: Yes

Reviewer #5: Yes

6. Review Comments to the Author

**Reviewer #1:**  The paper has improved in clarity by responding in depth to all the reviewers comments. However the "real wolrd data" presented through the FAERS data analysis does not really add anything new to the knowlege about solriamfetol

**Reviewer #3: ** I have no further comments. I believe, within the constraints of the methodology and dataset employed, this is the most that can be reasonably achieved.

Thank you.

**Reviewer #4:**  (No Response)

**Reviewer #5:**  As a clinician, I find it difficult to know exactly what to do with the data presented. Who is reporting in the FAERS and why? It is also not clear how this report on solriamfetol compares with reports on other similar drugs or drugs for similar indications.

For me, it is also not entirely clear from the current description exactly what kind of statistics were applied and how to fully understand them. But that is not crucial for the more general discussion of what to do with data from databases like this.

Apparently, most reports are from people who do not use solriamfetol for narcolepsy or OSAS. It then makes an addition, which is difficult for me to understand, that ‘the most frequently reported indication for solriamfetol was narcolepsy (596, 38.45%), followed by OSA (57, 3.68%)’. All other indications thus constitute < 3.68% of the total group. This implies that at least 15 other indications for solriamfetol were reported! Moreover, if I understand it correctly, the most often reported is that the drug is not effective. Personally, I would not interpret that as an AE.

Finally, methodological issues aside, what is reported on AEs is not really new or unexpected.

Nevertheless, the previous reviewer comments have been adequately addressed.

7. PLOS authors have the option to publish the peer review history of their article (what does this mean? ). If published, this will include your full peer review and any attached files.

**Do you want your identity to be public for this peer review?** For information about this choice, including consent withdrawal, please see our Privacy Policy .

Reviewer #1: No

Reviewer #3: No

Reviewer #4: **Yes: ** Markku Partinen

Reviewer #5: No

---

## [Author Response · Author response to Decision Letter 2]

31 Jul 2025

As a clinician, I find it difficult to know exactly what to do with the data presented. Who is reporting in the FAERS and why? It is also not clear how this report on solriamfetol compares with reports on other similar drugs or drugs for similar indications.

Response: We sincerely thank the reviewer for raising this important point. The primary objective of our study is to identify the most frequently reported AEs associated with solriamfetol based on data from the FAERS database. While we fully acknowledge that causality cannot be established from spontaneous reports, such analyses are widely used in pharmacovigilance to generate early safety signals and provide real-world insights that can inform clinical monitoring and future research.

Regarding the question of "who is reporting and why," we have clarified this in both the Methods section and Table 1, which provides reporter types. The majority of AE reports were submitted by healthcare professionals and consumers. We have now further emphasized this in the revised manuscript to improve clarity.

As for comparisons with other drugs used for similar indications (e.g., modafinil, pitolisant), we recognize the value of such analyses; however, our study was specifically designed to focus on solriamfetol and to explore its safety profile across all reported indications. Comparative analyses have been addressed in prior publications, and we consider them beyond the scope of the current investigation. We also have a brief summary in this article .

For me, it is also not entirely clear from the current description exactly what kind of statistics were applied and how to fully understand them. But that is not crucial for the more general discussion of what to do with data from databases like this.

Response: We appreciate the reviewer’s comment regarding the statistical methods used. In this study, we employed four widely accepted disproportionality analysis algorithms: Reporting Odds Ratio (ROR), Proportional Reporting Ratio (PRR), Bayesian Confidence Propagation Neural Network (BCPNN), and Multi-item Gamma Poisson Shrinker (MGPS).

Each method offers distinct advantages:

– ROR is effective in correcting biases that may arise from low event counts.

– PRR provides higher specificity in detecting safety signals.

– BCPNN excels in integrating multi-source data and performing cross-validation.

– MGPS is particularly useful for identifying rare adverse events.

By combining these four methods, we aimed to enhance signal detection robustness and reliability. The joint use of multiple algorithms allows for cross-validation, helping to reduce false positives and increase the credibility of the findings. This strategy also broadens the detection scope and allows for the identification of potential rare adverse reactions through threshold adjustments and variance control.

Apparently, most reports are from people who do not use solriamfetol for narcolepsy or OSAS. It then makes an addition, which is difficult for me to understand, that ‘the most frequently reported indication for solriamfetol was narcolepsy (596, 38.45%), followed by OSA (57, 3.68%)’. All other indications thus constitute < 3.68% of the total group. This implies that at least 15 other indications for solriamfetol were reported! Moreover, if I understand it correctly, the most often reported is that the drug is not effective. Personally, I would not interpret that as an AE.

Response: In response to the major concern regarding the heterogeneity of indications and lack of indication-specific analysis, we have added two new supplementary tables:

STable 2: Signal strength of adverse events associated with solriamfetol for the treatment of narcolepsy.

STable 3: Signal strength of adverse events associated with solriamfetol for the treatment of obstructive sleep apnea (OSA).

These tables allow for a clearer understanding of the safety profile of solriamfetol across its approved indications. The results show that while certain adverse events (e.g., drug ineffectiveness, headache, anxiety) are common to both indications, there are distinct signal differences in severity and ranking. We added a description of the corresponding content in the results section.

For reported indications. As shown in Table 1, we clearly listed the proportions of the two approved indications—narcolepsy (38.45%) and obstructive sleep apnea (3.68%)—in the included reports. However, 58.19% of the reports lacked specific indication information. This high percentage does not necessarily imply other indications. Rather, it reflects missing or incomplete data within the FAERS database, a common limitation in spontaneous reporting systems.

We agree that “drug ineffective” may not represent a physiological adverse event in the traditional sense. However, in pharmacovigilance practice, especially within spontaneous reporting systems like FAERS, “drug ineffective” is a commonly reported Preferred Term (PT) that reflects real-world dissatisfaction with therapeutic outcomes. While it does not necessarily indicate harm, a high frequency of such reports can serve as an important early signal for suboptimal drug response, inappropriate patient selection, dosing issues, or emerging patterns of therapeutic failure. Regulatory agencies such as the FDA consider these signals valuable for post-marketing surveillance and risk-benefit assessment.

Finally, methodological issues aside, what is reported on AEs is not really new or unexpected.

Response: We thank the reviewer for this important observation. In response to this concern, we have added a new paragraph to the Discussion section comparing the adverse events (AEs) identified in our study with those listed in the official U.S. prescribing information for solriamfetol (Sunosi®). While several commonly reported AEs such as headache, anxiety, and insomnia are consistent with the product labeling, we also identified a number of previously underreported or unlabeled AEs—including therapeutic response decreased, irritability, suicidal ideation, mania, and persecutory delusion—which showed statistically significant signals in our FAERS analysis.

Importantly, through indication-specific stratified analysis, we found that psychiatric-related AEs were disproportionately reported in patients with narcolepsy, whereas cardiovascular-related AEs were more prominent in patients with OSA. These findings go beyond the current labeling and highlight indication-specific safety concerns that may not have been captured in pre-marketing clinical trials. We believe this adds novel, clinically relevant insights and underscores the value of post-marketing pharmacovigilance in complementing existing drug safety information.

We hope that these major revisions sufficiently address the methodological and interpretational concerns raised and bring the manuscript in line with the standards of PLOS ONE.

Thanks for your attention to our manuscript!

Sincerely,

Kaijian Zhu

Email: 13515245847@163.com

---

## [Decision Letter · Decision Letter 2]

19 Aug 2025

PONE-D-25-14184R2Post-marketing Safety of Solriamfetol: A Retrospective Pharmacovigilance Study Based on the US Food and Drug Administration Adverse Event Reporting SystemPLOS ONE

Dear Dr. Zhu,

Thank you for submitting your manuscript to PLOS ONE. After careful consideration, we feel that it has merit but does not fully meet PLOS ONE’s publication criteria as it currently stands. Therefore, we invite you to submit a revised version of the manuscript that addresses the points raised during the review process. The manuscript is significantly better and should be published soon, but there are still errors in Table 2:

The first column of Table 2 is labeled “Preferred Terms” and the second column is labeled “System Organ Class”. There seam to be obvious errors here: e.g., the organ “Renal and urinary disorders” is mentioned for the preferred term “cataplexy”. What does cataplexy have to do with renal and urinary disorders?

Or Sleep attacks with neoplasms? On the other hand, some information is incomprehensible: for example, why does “Suicidal intention” is mentionned under "investigations"? Readers cannot understand this without a comment in the legend. 

The entire Table 2 appears to be full of errors. Please read carefully before resubmission all tables and ensure that the tables are error-free. 

We look forward to receiving your revised manuscript.

Kind regards,

Christian Veauthier, M.D.

Academic Editor

PLOS ONE

Journal Requirements:

Reviewers' comments:

Reviewer's Responses to Questions

**Comments to the Author**

1. If the authors have adequately addressed your comments raised in a previous round of review and you feel that this manuscript is now acceptable for publication, you may indicate that here to bypass the “Comments to the Author” section, enter your conflict of interest statement in the “Confidential to Editor” section, and submit your "Accept" recommendation.

Reviewer #3: All comments have been addressed

Reviewer #5: (No Response)

2. Is the manuscript technically sound, and do the data support the conclusions?

Reviewer #3: Yes

Reviewer #5: Yes

3. Has the statistical analysis been performed appropriately and rigorously? 

Reviewer #3: Yes

Reviewer #5: Yes

4. Have the authors made all data underlying the findings in their manuscript fully available?

Reviewer #3: Yes

Reviewer #5: (No Response)

5. Is the manuscript presented in an intelligible fashion and written in standard English?

Reviewer #3: Yes

Reviewer #5: Yes

6. Review Comments to the Author

Reviewer #3: Although this study inherently suffers from the limitations of utilizing FAERS as the data source, I do not think any additional revisions could improve the manuscript at this point. I do not have any further comments.

Reviewer #5: The previous comments have been properly addressed.

But, something went wrong in Table S2: the second column is not correctly matched to the first column. Whether there is possibly also an error in the other columns I cannot determine.

7. PLOS authors have the option to publish the peer review history of their article (what does this mean? ). If published, this will include your full peer review and any attached files.

**Do you want your identity to be public for this peer review?** For information about this choice, including consent withdrawal, please see our Privacy Policy .

Reviewer #3: No

Reviewer #5: No

---

## [Author Response · Author response to Decision Letter 3]

22 Aug 2025

22-Agu-2025

Dear Prof. Christian Veauthier,

Thank you for your time and valuable feedback on our manuscript entitled "Post-marketing Safety of Solriamfetol: A Retrospective Pharmacovigilance Study Based on the US Food and Drug Administration Adverse Event Reporting System" (Submission ID PONE-D-25-14184R2). We have carefully addressed all comments and suggestions. Modifications in the revised manuscript are highlighted in red, and point-by-point responses are provided below:

Additional Editor Comments:

The manuscript is significantly better and should be published soon, but there are still errors in Table 2:

The first column of Table 2 is labeled “Preferred Terms” and the second column is labeled “System Organ Class”. There seam to be obvious errors here: e.g., the organ “Renal and urinary disorders” is mentioned for the preferred term “cataplexy”. What does cataplexy have to do with renal and urinary disorders?

Or Sleep attacks with neoplasms? On the other hand, some information is incomprehensible: for example, why does “Suicidal intention” is mentionned under "investigations"? Readers cannot understand this without a comment in the legend.

The entire Table 2 appears to be full of errors. Please read carefully before resubmission all tables and ensure that the tables are error-free.

Reviewer: 5

Comment 1

The previous comments have been properly addressed.

But, something went wrong in Table S2: the second column is not correctly matched to the first column. Whether there is possibly also an error in the other columns I cannot determine.

Response to Editor and Reviewer #5:

Thank you for your valuable feedback. We have carefully reviewed your comments and thoroughly re-examined the manuscript. Upon our careful recheck, we found that the issue was primarily confined to Table S2, where there was a misalignment between the "Preferred Terms" and the "System Organ Classes." This error was due to incorrect matching during the table preparation process, which led to the confusion, such as the association of "cataplexy" with "renal and urinary disorders" and "sleep attacks" with "neoplasms."

We have corrected the entries in Table S2 to ensure that the "Preferred Terms" are properly aligned with their corresponding "System Organ Classes." After reviewing the rest of the tables, we found no similar issues.

We believe these revisions have significantly improved the accuracy of the tables, and we have attached the revised manuscript with all necessary corrections. We hope these adjustments meet your expectations.

Thanks for your attention to our manuscript!

Sincerely,

Kaijian Zhu

Email: 13515245847@163.com

---

## [Editor Report · Decision Letter 3]

3 Sep 2025

PONE-D-25-14184R3Post-marketing Safety of Solriamfetol: A Retrospective Pharmacovigilance Study Based on the US Food and Drug Administration Adverse Event Reporting SystemPLOS ONE

Dear Dr. Zhu,

Thank you for submitting your manuscript to PLOS ONE. After careful consideration, we feel that it has merit but does not fully meet PLOS ONE’s publication criteria as it currently stands. Therefore, we invite you to submit a revised version of the manuscript that addresses the points raised during the review process.

Unfortunately, I have one more comment to make: 

in Table 2, the preferred term “sleepiness” is assigned to system class “psychiatric disorder.” 

I find this difficult to understand given that the article is about narcolepsy and daytime sleepiness, 

and it reveals that the authors have no concept of daytime sleepiness and think that sleepiness is psychologically induced. That is not correct: Sleep apnea can cause sleepiness. Narcolepsy can cause sleepiness as well: neither is a psychiatric disorder. Sleepiness per se is not psychiatric. Please change the organ class.

We look forward to receiving your revised manuscript.

Kind regards,

Christian Veauthier, M.D.

Academic Editor

PLOS ONE
---

## [Author Response · Author response to Decision Letter 4]

7 Sep 2025

7-Sep-2025

Dear Prof. Christian Veauthier,

Thank you for your time and valuable feedback on our manuscript entitled "Post-marketing Safety of Solriamfetol: A Retrospective Pharmacovigilance Study Based on the US Food and Drug Administration Adverse Event Reporting System" (Submission ID PONE-D-25-14184R3). We have carefully addressed all comments and suggestions. Modifications in the revised manuscript are highlighted in red, and point-by-point responses are provided below:

Editor Comments:

Unfortunately, I have one more comment to make:

in Table 2, the preferred term “sleepiness” is assigned to system class “psychiatric disorder.”

I find this difficult to understand given that the article is about narcolepsy and daytime sleepiness,

and it reveals that the authors have no concept of daytime sleepiness and think that sleepiness is psychologically induced. That is not correct: Sleep apnea can cause sleepiness. Narcolepsy can cause sleepiness as well: neither is a psychiatric disorder. Sleepiness per se is not psychiatric. Please change the organ class.

Response: We sincerely thank the editor for pointing out this important issue. We fully agree that sleepiness is not a psychiatric disorder, and conditions such as narcolepsy or obstructive sleep apnea can lead to excessive sleepiness through non-psychiatric mechanisms. I will now provide a detailed explanation.

We carefully reviewed our data and would like to clarify that in our Table 2, only the SOCs are presented, and the Preferred Term “sleepiness” does not actually appear in our tables. All adverse events were coded strictly according to the standardized MedDRA dictionary, which automatically maps PT to predefined SOCs. In MedDRA, some terms related to excessive daytime sleepiness (e.g., sleep attacks) may be automatically classified under the SOC “psychiatric disorders,” even though clinically these events are primarily neurological or sleep–wake phenomena rather than psychiatric in nature. This point has already been introduced in the Methods section. Therefore, the classification reflects the MedDRA structure rather than our interpretation.

In our analysis, we first presented the results at the SOC level to give a global overview of the organ systems most frequently involved. However, as the editor correctly pointed out, the SOC classification is based on MedDRA conventions and may sometimes group sleep-related events (e.g., sleep attacks) under “psychiatric disorders,” even though clinically they are not psychiatric in nature. To address this limitation and provide readers with a more accurate understanding, we also performed a PT–level analysis and explicitly listed the PTs together with their assigned SOCs in MedDRA structure. By doing so, readers can clearly see the specific events that contributed to each SOC signal and avoid misinterpreting sleep-related symptoms as psychiatric disorders.

In the revised manuscript, we have added clarifications in the Discussion to emphasize that SOC assignment follows MedDRA rules, while the PT-level analysis allows a more nuanced interpretation in line with clinical reality. We believe this approach enhances transparency and directly addresses the editor’s concern.

We sincerely thank the editor again for highlighting this important issue, which has helped us improve the clarity and rigor of our manuscript.

Thanks for your attention to our manuscript!

Sincerely,

Kaijian Zhu

Email: 13515245847@163.com

---

## [Editor Report · Decision Letter 4]

10 Sep 2025

Post-marketing Safety of Solriamfetol: A Retrospective Pharmacovigilance Study Based on the US Food and Drug Administration Adverse Event Reporting System

PONE-D-25-14184R4

Dear Dr. Zhu,

We’re pleased to inform you that your manuscript has been judged scientifically suitable for publication and will be formally accepted for publication once it meets all outstanding technical requirements.

Kind regards,

Christian Veauthier, M.D.

Academic Editor

PLOS ONE
---

## [Editor Report · Acceptance letter]

PONE-D-25-14184R4

PLOS ONE

Dear Dr. Zhu,

I'm pleased to inform you that your manuscript has been deemed suitable for publication in PLOS ONE. Congratulations! Your manuscript is now being handed over to our production team.

Kind regards,

on behalf of

Dr. Christian Veauthier

Academic Editor

PLOS ONE